# Micro-Pulse Lidar Cruising Measurements in Northern South China Sea

**Yuan Li** [1,2]**, Baomin Wang** [1,2] **, Shao-Yi Lee** [1,2]**, Zhijie Zhang** [1]**, Ye Wang** [1] **and Wenjie Dong** [1,2,*]

[1] School of Atmospheric Sciences and Guangdong Province Key Laboratory for Climate Change and Natural Disaster Studies, Sun Yat-sen University, Zhuhai 519082, China; liyuan75@mail2.sysu.edu.cn (Y.L.); wangbm@mail.sysu.edu.cn (B.W.); lishaoyi@mail.sysu.edu.cn (S.-Y.L.); zhzhij33@mail2.sysu.edu.cn (Z.Z.); wangy393@mail2.sysu.edu.cn (Y.W.)

[2] Southern Marine Science and Engineering Guangdong Laboratory (Zhuhai), Zhuhai 519082, China

[*] Correspondence: dongwj3@mail.sysu.edu.cn

**Abstract:** A shipborne micro-pulse lidar (Sigma Space Mini-MPL) was used to measure aerosol extinction coefficient over the northern region of the South China Sea from 9 August to 7 September 2016, the first time a mini-MPL was used for aerosol observation over the cruise region. The goal of the experiment was to investigate if the compact and affordable mini-MPL was usable for aerosol observation over this region. The measurements were used to calculate vertical profiles of volume extinction coefficient, depolarization ratio, and atmospheric boundary layer height. Aerosol optical depth (AOD) was lower over the southwest side of the cruise region, compared to the northeast side. Most attenuation occurred below 3.5 km, and maximum extinction values over coastal areas were generally about double of values offshore. The extinction coefficients at 532 nm (aerosol and molecular combined) over coastal and offshore areas were on average 0.04 km$^{-1}$ and 0.02 km$^{-1}$, respectively. Maximum values reached 0.2 km$^{-1}$ and 0.14 km$^{-1}$, respectively. Vertical profiles and back-trajectory calculations indicated vertical and horizontal layering of aerosols from different terrestrial sources. The mean volume depolarization ratio of the aerosols along the cruise was 0.04. The mean atmospheric boundary layer height along the cruise was 653 m, with a diurnal cycle reaching its mean maximum of 1041 m at 12:00 local time, and its mean minimum of 450 m at 20:00 local time. Unfortunately, only 11% of the measurements were usable. This was due to ship instability in rough cruise conditions, lack of stabilization rig, water condensation attached to the eye lens, and high humidity attenuating the echo signal. We recommend against the use of the mini-MPL in this cruise region unless substantial improvements are made to the default setup, e.g., instrument stabilization, instrument protection cover, and more theoretical work taking into account atmospheric gas scattering or absorption.

**Keywords:** South China Sea; marine aerosol; lidar; cruising observation; extinction coefficient; depolarization ratio; atmospheric boundary layer height

## 1. Introduction

Aerosols play an important role in the constitution of the troposphere. Due to their effectiveness in absorbing and scattering solar radiation, aerosols can alter atmospheric temperature stratification and planetary albedo. As cloud condensation nuclei, they are indispensable in the water cycle. Changes in aerosol distributions can change the Earth's energy balance to alter regional and global climates [1–5]. As the largest constituent of natural aerosols, marine aerosols account for 44% of the global aerosols [6,7]. Unfortunately, the study of marine aerosols began later compared to that of terrestrial aerosols, due to

limitation by technology and instruments. Marine aerosol research has developed rapidly in recent years with the advancement of technology [8–12].

The measurement of marine aerosols is needed to quantify their aerosol radiative effects and their impact on regional climate. For example, previous studies have confirmed the influence of typhoons on aerosol concentrations, and while aerosol concentrations have little impact on the storm track, they in turn influence the structure and intensity of typhoons [13,14]. Unfortunately, much of the typhoon-aerosol interaction takes place over the South China Sea, where actual observations are lacking. This motivated the authors to perform aerosol measurements during Sun Yat-sen University's first research cruise in the northern South China Sea during the summer typhoon season, with the hope of obtaining some aerosol measurements near a typhoon. However, the main aim of the first cruise was a feasibility experiment for the instrumental setup, focusing on the affordable and compact mini-MPL.

The South China Sea is the third largest inland sea in the world, with unique geographical and climatic conditions. Rainfall is abundant over the South China Sea and there is a strong East Asian monsoon signal [15,16]. It is also one of the major areas of typhoon activity [17]. Increased emission of industrial aerosols in recent years have altered the inland horizontal and vertical distribution of aerosols [18], but the extent to which offshore aerosol distributions have been affected is not clear.

The main components of marine aerosols are sea salt, sulfates, and other secondary aerosols formed in atmosphere-ocean exchanges. Existing measurement methods include multi-spectral means, polarization combined multi-angle detection methods, and lidar detection. There have been a number of studies on marine aerosol optical properties over the north South China Sea using solar photometers [19–21], but there are no lidar studies of marine aerosols over the South China Sea that the authors know of in publicly accessible scientific literature.

Lidar is an active remote sensing device that emits a laser light beam to receive aerosol backscatter, hence obtaining information on aerosol vertical distribution [22]. The Micro-pulse Lidar (MPL) is a sophisticated laser remote sensing system able to profile atmospheric cloud and aerosol scattering, and hence useful in scientific studies and environmental monitoring. The MPL measures the intensity of backscattered light after transmitting a laser pulse and then transforms the signal into atmospheric information in real time. It has been successfully used in oceanic observations over other regions in recent years [23]. Welton et al. carried out cruising observations in the Indian Ocean and found the aerosols there to be concentrated within 4 km of surface [24]. Li et al. measured the aerosol optical thickness over the Yellow Sea to be about 0.1, with weak diurnal and day-to-day variation [25]. Duflot et al. measured the average backscatter-to-extinction ratio of a layer of marine aerosols in the southern Indian Ocean to be 26 ± 6 sr [26].

The atmospheric boundary layer (ABL), also known as the planetary boundary layer, plays an important role in the transport of not just momentum, moisture, and heat in the troposphere but also aerosols. The atmospheric boundary layer height (ABLH) is the height at which the turbulent motions from beneath are unable to penetrate beyond [27–29]. For the well-mixed ABL, aerosol particles are well mixed by the thermally driven turbulence, and the height of mixing layer is more or less identical to the ABLH [30]. The atmospheric boundary layer height cannot currently be directly measured but can be derived from the vertical profiles of meteorological variables or lidar echo signal. Lavers et al. in 2019 used data from Integrated Forecast System of the European Centre for Medium-Range Weather Forecasts to calculate the average observed ABLH over the ocean, obtaining a value of 744.2 m [31,32].

The authors decided to use a micro-pulse lidar (mini-MPL) in this study, to see if a different instrument can reveal any new information on marine aerosols in the northern South China Sea. For example, in this study the atmospheric extinction coefficient was calculated from the echo signal using the Fernald method (Section 2.3), while the ABLH was inferred from the gradient of the echo signal using the gradient method (Section 2.5). The attractiveness of the mini-MPL lies in its affordability and compactness. To examine the feasibility of using this instrument, the aerosol spectrum of the aerial light in the northern part of the South China Sea was observed during the cruise period by using the shipborne mini-MPL. The cruise experiment and inversion methodology will be presented in Section 2.

The results are shown in Section 3. Section 4 will discuss the results obtained. The conclusions of the study will be summarized in Section 5.

## 2. Observation Experiments and Methodology

### 2.1. Cruise and Instrument

The research vessel "Sea Tune 6", a refurbished 1000-tonne fishing vessel, departed from Tunshun, Zhanjiang, on 8 August 2016, and followed the cruise route shown in Figure 1 until it reached Nansha, Guangzhou, on 8 September 2016. Measurements were taken during the period from 9 August to 7 September 2016, during both day and night.

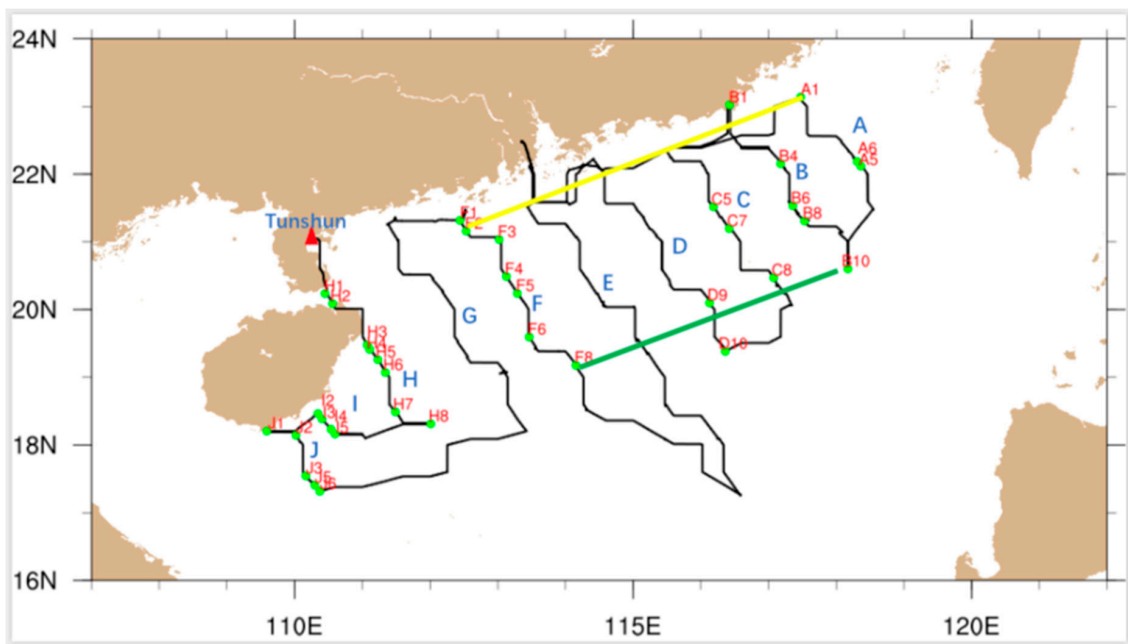

**Figure 1.** The path of the research cruise (black line). Green dots mark the positions of usable measurements after quality control, and their station names are labeled in red text with A-J indicating the cruise section. Stations close to the yellow line were considered near-shore, while stations close to the green line were considered off-shore.

The micro-pulse lidar used for observation was manufactured by Sigma Space (Mini-MPL). The system consists of three parts: The transmitter system, the receiving system, and the acquisition system. The host and the telescope are connected by optic fiber. The whole system adopts a coaxial design. The emission system uses Nd:YAG lasers (neodymium-doped yttrium aluminium garnet) with a central wavelength of 532 nm, a laser pulse repetition frequency of 2.5 kHz, a single pulse of energy of 3.6 μJ, a spatial resolution of 15 m, and a time resolution of 5 s. The time resolution we used for the retrieval of optical properties was 5 min. The maximum detection range was 10 km during daytime and 15 km during night. The minimum detection range was less than 0.1 km [33]. During the cruise, the MPL system on the ship made measurements every day from from 0.3 km to 4 km. The key advantage of the Mini-MPL is its portability, allowing it to be easily installed on the research vessel.

### 2.2. Quality Control and Preprocessing

The cruise was divided into 10 cruise sections and measurements were taken continuously along the cruise path. Due to the experimental nature of the setup and the experimenters' lack of familiarity with the marine conditions of the cruise region, the measurements over many sections of the cruise

were of suspect quality. The following quality control process was carried out by visually inspecting every profile, excluding periods of measurements when:

- It was discontinuous due to technical difficulties, e.g., ship instability or typhoon avoidance.
- There were 10-m wind speed above 9 m/s or when it was raining. From cruise weather records, the profile was judged to be visibly contaminated by clouds or excessive humidity. The profile was considered contaminated by excessive humidity when both visibility was less than 7 km and the signal-to-noise ratio was less than 20 below 3 km.
- The raw signal was overshooting.

After quality control, only about 11% of all raw measurements were usable. These were along the eight sections shown in Figure 1A–J. There were no usable measurements along section E and section G. The authors decided against further quality control as most of the raw measurements had already been excluded and attempted to process the remaining measurements "as-if" they were good. The authors believe that ship instability constituted the major source of uncertainty in the measurement, e.g. from large swells, and particularly when typhoons were in the region.

Note that the measurements that passed quality control were pre-processed by the Sigma Space software before inversion. This pre-processing included detector no-load time correction, background noise correction, post-pulse correction, overlap factor correction, distance compensation correction, distance corrections, etc. For example, the overlap factor correction compensated for signal losses when the return signal could not be fully accepted at short ranges, due to optical reasons. This resulted in the returned signal falling outside the receiving field of view, so an overlap factor correction was required. The overlap function $O_c(Z)$ can be written as

$$P(Z) = O_c(Z)P_r(Z) \tag{1}$$

where $P(Z)$ is the corrected value of the echo signal $P_r(Z)$ returned from height $Z$. For the mini-MPL, the transmitting field of view was 50 μrad while the receiving field angle was 100 μrad [34].

All data discussed below underwent the above-described pre-processing.

### 2.3. Calculation of Extinction Coefficient

The aerosol extinction coefficient represents the reduction of radiation in a band due to scattering and absorption by aerosols. The following inversion calculation of aerosol extinction coefficient from the lidar signal assumes that profiles with low clouds and excessive water vapor have been removed.

The lidar equation can be written as:

$$P(Z) = C\beta(Z)exp\left[-2\int_0^z \sigma(Z')dZ'\right] \cdot Z^{-2} \tag{2}$$

where $P(Z)$ is the echo signal returned by the $Z$ height, C is the lidar constant, $\beta(Z)$ is the 180° backscattering cross-section, and $\sigma(Z)$ is the extinction coefficient. The extinction coefficient consists of absorbing and scattering components, $\sigma = \sigma^{abs} + \sigma^{sca}$. The backscatter coefficient is related through $\beta = \Theta(\pi)\sigma^{sca}$, where $\Theta(\theta)$ is the scattering phase function at angle $\theta$. To solve for the two unknown quantities, it is typical to determine some value of $s = \sigma/\beta$, the lidar ratio.

Methods commonly used to solve Equation (2) are the Collis slope, Klett, and Fernald methods [35–37]. In this study, the Fernald method was used to calculate the extinction coefficient.

The Fernald method divides the atmosphere into two parts: air molecules and aerosols. The extinction coefficient $\sigma$ and the backscattering coefficient β are accordingly considered as consisting of two contributions. Hence, Equation (2) can be written as Equation (3), with subscripts $p$ and $m$ referring to contributions by aerosols and air molecules, respectively:

$$P(Z) = C\left(\beta_p(Z) + \beta_m(Z)\right)exp\left\{-2\int_0^Z \left[\sigma_p(Z') + \sigma_m(Z')\right]dZ'\right\} \cdot Z^{-2} \tag{3}$$

P(Z) is measured by the lidar. The lidar constant C is known and unique to the lidar system. The extinction coefficient and backscattering coefficient of atmosphere at each height, $\sigma_m(Z)$ and $\beta_m(Z)$, can be calculated using Rayleigh scattering theory from vertical density, assuming the American standard atmosphere, as is commonly done. For the atmosphere, $s_m = \frac{\sigma_m}{\beta_m} = 8\pi/3$.

To solve Equation (3), let $X(Z) = P(Z)Z^2$ and perform forward and backward integrations of $X(Z)$ from $Z_0$, a calibration height of 6.0 km in this study, the default provided by the lidar's manufacturer. This gives Equations (4) and (5), providing $\sigma_p$ at heights $Z$ above and below $Z_0$.

$$\sigma_p(Z) = -\frac{s_p}{s_m} \cdot \sigma_m(Z) + \frac{X(Z) \cdot exp[2\left(\frac{s_p}{s_m} - 1\right) \int_Z^{Z_0} \sigma_m(Z')dZ']}{\frac{X(Z_0)}{\sigma_p(Z_a) + \frac{s_p}{s_m}\sigma_p(Z_0)} + 2\int_Z^{Z_a} X(Z')exp[2\left(\frac{s_p}{s_m} - 1\right)\int_Z^{Z_0} \sigma_m(Z'')dZ'']dZ'} \tag{4}$$

$$\sigma_p(Z) = -\frac{s_p}{s_m} \cdot \sigma_m(Z) + \frac{X(Z) \cdot exp[-2\left(\frac{s_a}{s_b} - 1\right) \int_{Z_0}^{Z} \sigma_b(Z')dZ']}{\frac{X(Z_a)}{\sigma_p(Z_a) + \frac{s_p}{s_m}\sigma_m(Z_a)} - 2\int_{Z_0}^{Z} X(Z')exp[-2\left(\frac{s_p}{s_m} - 1\right)\int_{Z_0}^{Z} \sigma_m(Z'')dZ'']dZ'} \tag{5}$$

From Equations (4) and (5), $\sigma_p$ and $s_p$ can be calculated. Hence, if $s_p$ is known, we can calculate $\sigma_p$. In practice, $s_p$ should be determined with the aid of a solar photometer [38]. However, this instrument was not present on the cruise. A value of $s_p = 30$ sr was chosen, the default provided by the manufacturer. This value was consistent with prior work over the coastal regions near the cruise path, which yielded values of 30–50 sr (Zhang, personal communication). A review of existing literature indicated a similar range of $s_p$ values: Cattral et al. also found a lidar ratio of 28 ± 5 sr over ocean [39]; Omar et al. found a lidar ratio ranging from 20 to 40 sr [40]; He et al. reported the mean lidar ratio was 29 ± 5.8 sr with minimum of 18 sr and maximum of 44 sr on the Hong Kong coast [41]; Muller et al. retrieved lidar ratio in the Indian Ocean ranging from 35 to 55 sr for mixture of clean marine and continental aerosol and 20 to 30 sr for clean marine condition at 532 nm [42]; Dawson found the mean global $s_p$ of marine aerosol was 26 sr [43].

Sensitivity testing with a range of lidar ratios should typically be done to determine the uncertainty of $\sigma_p$. This was not carried out for the collected data because of the unsatisfactory quantity/quality of measurements. The authors believe that the uncertainty caused by synoptic conditions exceeded the uncertainty due to the choice of $\sigma_p$. To estimated uncertainty in retrieved $\sigma_p$, an artificial signal based on coastal measurements in the region was processed with lidar ratio $s_p$ varying from 25 to 50, producing an uncertainty of 7–15%.

### 2.4. Calculation of Depolarization Ratio

The mini-MPL system uses scattering of polarized light to distinguish between spherical and nonspherical particles, in order to guess at the particle species. The linear volume depolarization ratio δ is the ratio of the backscatter intensities perpendicular to ($\beta_\perp$) and parallel to ($\beta_\parallel$) the outgoing lidar polarization:

$$\delta = \frac{\beta_\perp}{\beta_\parallel} \tag{6}$$

To avoid the background noise generated by sunlight during the day, and the influence of cloud scattering on the transmitted and the echo signals, only data recorded at night and conditions without low clouds were selected for the analysis of the depolarization ratio. The volume depolarization ratio is presented in this study, the particle depolarization ratio was not calculated due to technical reasons.

### 2.5. Calculation of Atmospheric Boundary Layer Height (ABLH)

There are many methods for calculating the ABLH, including the wavelet transform, Richardson number, and gradient methods [44–46], the last of which was used in this study. The gradient method is based on the effect of thermal inversion, which results in a large change in aerosol concentration

between the boundary layer (high) and the free atmosphere (low). The ABLH is determined based on the vertical gradient changes in aerosol concentration. As the lidar backscattered echo signal is positively correlated with the aerosol particle mass concentration, the gradient of the backscattered echo signal can be used instead. The distance-square correction of the meter-scattered echo signals collected by the lidar:

$$P(Z)Z^2 = CY(Z)\beta(Z)T(Z) \tag{7}$$

where $P(Z)$ is the echo signal returned by the Z height, C is the lidar system constant, $Y(Z)$ is overlap factor, $\beta(Z)$ is the backscattering coefficient, and $T(Z)$ is atmospheric transmissivity.

$$D(Z) = d\left[P(Z)Z^2\right]/dZ \tag{8}$$

when $D(Z)$ reaches its minimum, the value of the height is the ABLH.

*2.6. Back-Trajectories*

The Hybrid Single Particle Lagrangian Integrated Trajectory Model (HYSPLIT) model produced by the United States National Oceanic and Atmospheric Administration (NOAA) Air Resources Laboratory (ARL) was used to calculate backward trajectories of air parcels from measurement locations and times [47]. Meteorological driving was provided by NOAA's Global Data Assimilation System (GDAS) at 0.5° resolution (GDAS0p5). The model was run interactively online on the HYSPLIT READY (Real-time Evironmental Application and Display sYstem) website (https://www.ready.noaa.gov/HYSPLIT.php) [48]. Air parcels were initiated at altitudes of 200 m, 700 m, and 1300–1500 m, depending on the altitudes of maxima in the vertical profiles of extinction coefficient. Note that while the back-trajectories were calculated for 72 h, a trajectory can be considered "extinguished" upon reaching the surface, which is 0 m over the sea, but below 500–1000 m when crossing high topography into continental Southeast Asia. In addition to these, the effect of perturbations in location and initiation time were tested using the "ensemble" and "frequency" options of the READY website.

## 3. Results

*3.1. Vertical Structure of Extinction Coefficient*

The vertical profiles of mean aerosol extinction coefficient along each cruise section are shown in Figure 2. Note that the number of usable measurements differed for each section (Figure 1) and all the signals chosen were during sunny day in order to remove the effect of clouds and other complicated synoptic situations (e.g., rain, fog), and all profiles with low cloud have been removed by visual inspection, although practically it was impossible to exclude all interfering situations. Furthermore, the extinction coefficients presented in the following discussion were calculated with a constant value of $s_p = 30$ sr for all heights, rather than $s_p(Z)$, which adds another level of uncertainty to the values calculated.

The extinction coefficients below 1 km and 1–2 km were 0.02–0.07 km$^{-1}$ and 0.009–0.05 km$^{-1}$, respectively. From there, extinction coefficients declined unevenly but were effectively 0 km$^{-1}$ by 3.5 km. Cumulatively, more than 40% and 73% of total attenuation occurred below 1 km and 2 km, respectively. Although the pre-processing software had carried out some overlap correction with usable signals beginning from 255 m, the profiles of high extinction coefficients near the surface suggest that the correction could be improved.

Under conditions without low clouds, assuming aerosols to be solely responsible for the attenuation, aerosols were mainly concentrated below about 2 km, and aerosol levels were very low above about 3.5 km. The range of extinction coefficient and low extinction coefficient above 3.5 km was similar to the findings of Duflot et al. over the southern Indian Ocean using both lidar and solar photometer and the vertical distribution was also similar [26]. Powel et al. measured the oceanic aerosol extinction by using MPL lidar and yielded aerosol extinctions ranging from 0.02 to 0.07 km$^{-1}$ for clean conditions [23],

which is close to the range from section H to section J that was in the southwest of the cruise region. Wang et. al reported oceanic aerosol of $0.079 \pm 0.001$ km$^{-1}$ before a dust storm occurred in the northern South China Sea [49]. Compared to the land, where the extinction coefficient of the aerosol in the land is typically $0.1$–$0.5$ km$^{-1}$, the marine aerosol is clearer than land.

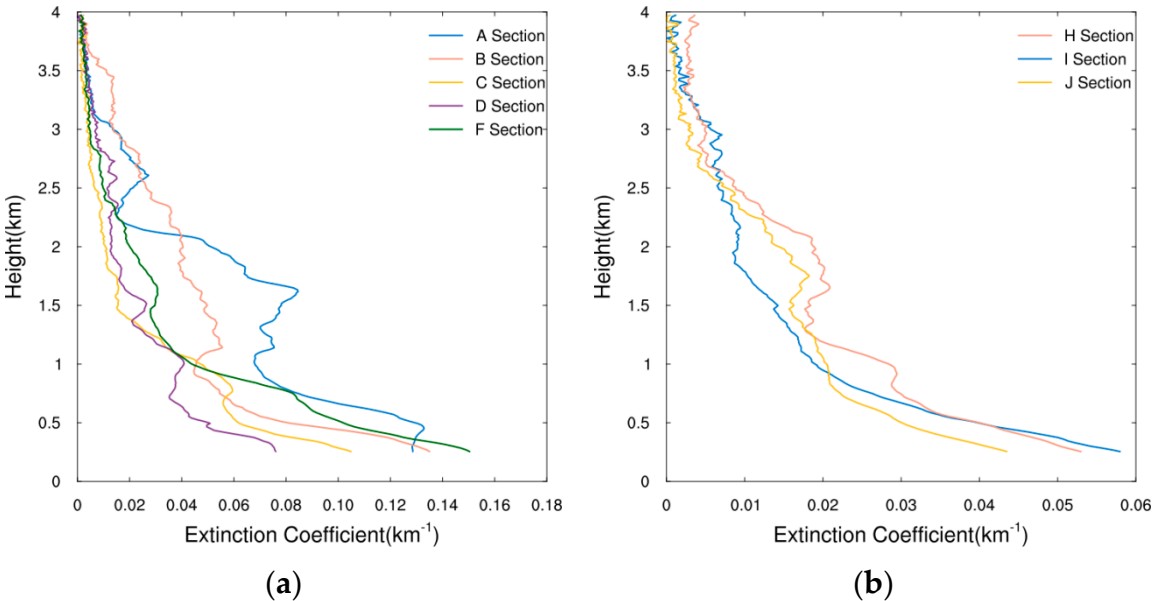

**Figure 2.** Vertical profiles of the extinction coefficient averaged along (**a**) sections A, B, C, D, F, and (**b**) sections H, I, J. There were no usable measurements along sections E and G. Note that the scale along the abscissa in panel (**b**) is one-third of that in panel (**a**).

The eight mean profiles fell naturally into two groups, sections A–F (Figure 2a) and sections H–J (Figure 2b), with higher and lower extinction coefficients below 2 km, respectively. The lowest values of extinction coefficient in the first group, found in profiles C and D, were higher or similar in value to the highest values in the second group. These sections fell geographically into the northeast and southwest sides of the cruise region (Figure 1), which were downstream and upstream based on the mean seasonal low-level winds over the region (Figure 3, data is from ERA-Interim [50]). However, extinction coefficients did not increase monotonically in either direction along the coast. For example, the largest values in the northeast group were found on the northeast-most profile (A) at 0.5–2 km, followed by values from profile B at 1–2 km, but profile F was at 0.5–1 km.

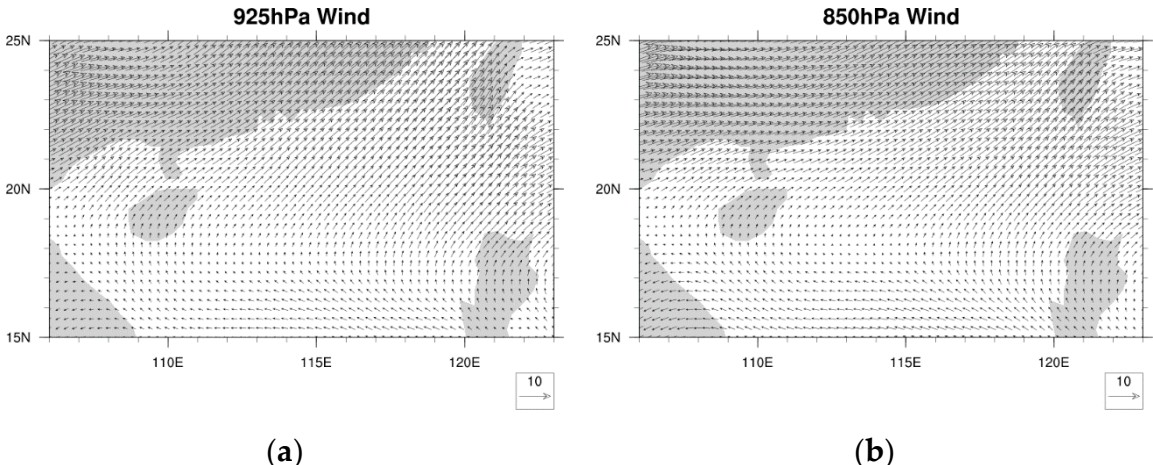

**Figure 3.** Daily mean wind fields of the cruising period at (**a**) 925 hPa and (**b**) 850 hPa.

An alternative way to group the stations was according to their distance away from the shore, since stations closer to the shore generally (but not always) show higher extinction coefficients below 0.5 km compared to stations further away from the shore. We defined four cross-sections (CS) for the stations in the northeast region: CS1 (A1, B1, F1, F2), CS2 (B4, C5, F3, F4, F5), CS3 (A5, A6, B6, C7, B8, F6), and CS4 (F8, D9, D10, C8, B10). CS1 and CS4 were considered to be near-shore and off-shore profiles, marked in Figure 1 as yellow and green lines, respectively. CS2 and CS3 were considered intermediate profiles (not marked to maintain the clarity of Figure 1). Stations in the southeast region (H, I, J) were excluded, to avoid complications from the curving shoreline and Hainan Island.

The vertical profiles of mean extinction coefficient along the four cross-sections are shown in Figure 4. Extinction coefficients below 1 km for on-shore and off-shore profiles were 0.06–0.18 km$^{-1}$ and 0.04–0.09 km$^{-1}$, respectively. The extinction coefficients between 1 to 2 km were 0.03–0.06 km$^{-1}$ and 0.02–0.03 km$^{-1}$, respectively. Cumulatively, more than 83% and 76% of total attenuation occurred below 2 km in near-shore and off-shore profiles, respectively. Hence, while aerosols were concentrated below 2 km, both near-shore and off-shore, aerosol concentrations were about two times higher near-shore compared to off-shore. The values of extinction coefficient on the on-shore profile was higher or similar to values on the off-shore profile at all altitudes. Considering intermediate profiles as well, extinction coefficients reached similar values around 3 km, suggesting off-shore transport of terrestrial material up to this vertical limit (c.f., Section 3.2). However, again extinction coefficients did not decrease monotonically with distance from the coast. While values were similar between the two intermediate profiles of CS2 and CS3 below 1 km, values on CS3 were higher at 1–3 km despite the stations being further from the coast.

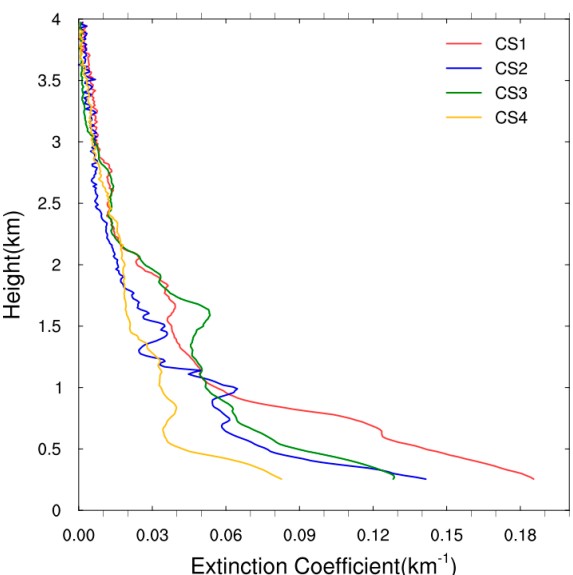

**Figure 4.** Vertical profiles of the extinction coefficient averaged by distance from the shore, in cross-section (CS) groups CS1, CS2, CS3, CS4. Stations in CS1 were closest to shore, while those in CS4 were furthest from the shore.

Both Figures 2 and 4 reveal the presence of one or even two layers of aerosols between 0.5 and 3 km, such as in the profiles of Section A (Figure 2), CS1, and CS2 (Figure 4). The vertical profiles of extinction coefficient for individual stations are shown in Figure 5, grouped by profile. The existence of multiple layers at 0.5–1.0 km and 1.0–1.5 km was particularly distinct at the three A- stations, B4, B10, C5, and F6. A weak layer at 1.5–2.0 km may have been present at H4-H8. These individual station plots supported our earlier conclusions that below 0.5 km, aerosol concentrations did not monotonically drop further away from the coast. This offshore low-level layer of aerosol was particularly distinct at B10 and F5-F6, and to a lesser extent at D10, H8, I4-I5, and J6. While these stations were approximately

upstream and downstream of one another based on the mean seasonal low-level winds (Figure 3), the authors could not determine any best way to make cross-sections that could capture the structure of all three layers.

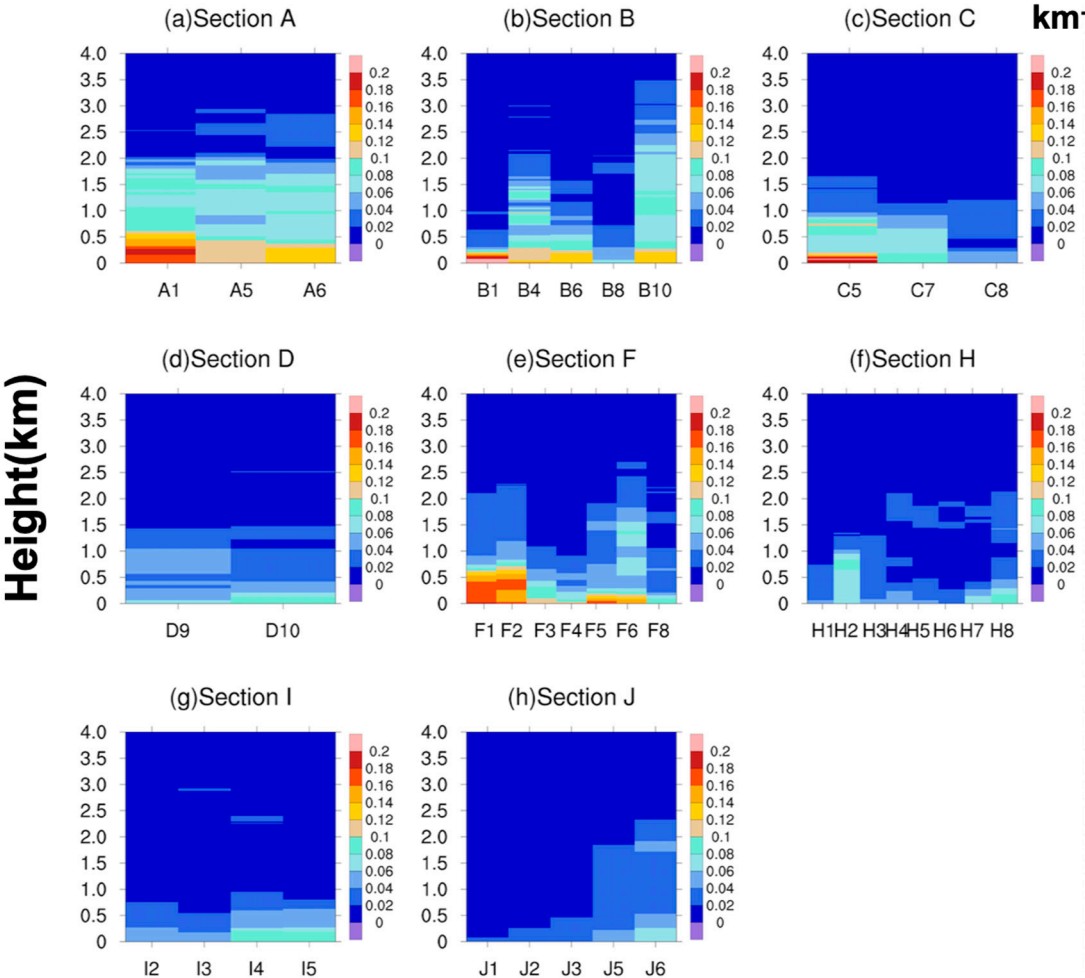

**Figure 5.** Vertical profiles of extinction coefficient at individual stations, grouped by cruise section: (**a**–**d**) Sections A to D, (**e**–**h**) sections F–J. There were no usable measurements along sections E and G.

### 3.2. Depolarization Ratio

The vertical profiles of depolarization ratio observed in each section were similar. Figure 6 shows the mean night-time depolarization ratio from 12 August to 25 August. The depolarization ratio averaged over the entire profile was 0.042 (0.039–0.046). Marine aerosols typically have depolarization ratios of 0.02 ± 0.02 due to their almost spherical shape [27]. The depolarization ratio of aerosols below 4 km was well within this range, indicating that the bulk of the measured aerosols were still marine in nature. Within this range, three layers of higher depolarization ratio could be seen, nearest to the surface, around 1 km, and around 2 km, supporting the presence of aerosol transport from terrestrial sources. The depolarization ratio rose above 4 km to about 0.08 at 10 km.

### 3.3. Atmospheric Boundary Layer Height (ABLH)

The atmospheric boundary layer height (ABLH) calculated during the cruise is shown in Figure 7. Note that multiple values of ABLH were measured at each station. If several layers of aerosol were found when determining the ABLH, the lowest layer was used. The mean ABLH during the cruise was 653.2 m. The maximum ABLH was 1693.8 m over station D9. The minimum ABLH was 219.3 m over station C5. The median values and 25th–75th percentile ranges of ABLH did not differ too much

from station to station. The diurnal cycle of ABLH is shown in Figure 8. The maximum and minimum hourly mean ABLH occurred at 12:00 (1041.2 m) and 20:00 (450.0 m), respectively. The relatively small station-to-station and diurnal range of the marine ABLH was expected, due to its relative stability compared to continental boundary layers. There was a large variability in the extreme values of ABLH, with values in the upper range close to typical continental values.

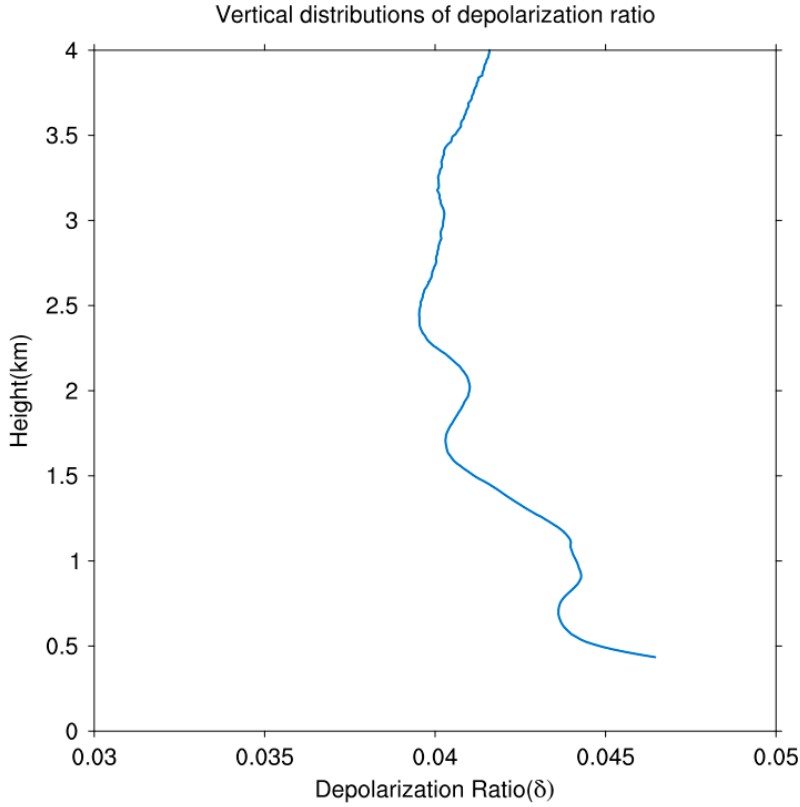

**Figure 6.** The vertical profile of mean depolarization ratio over all stations.

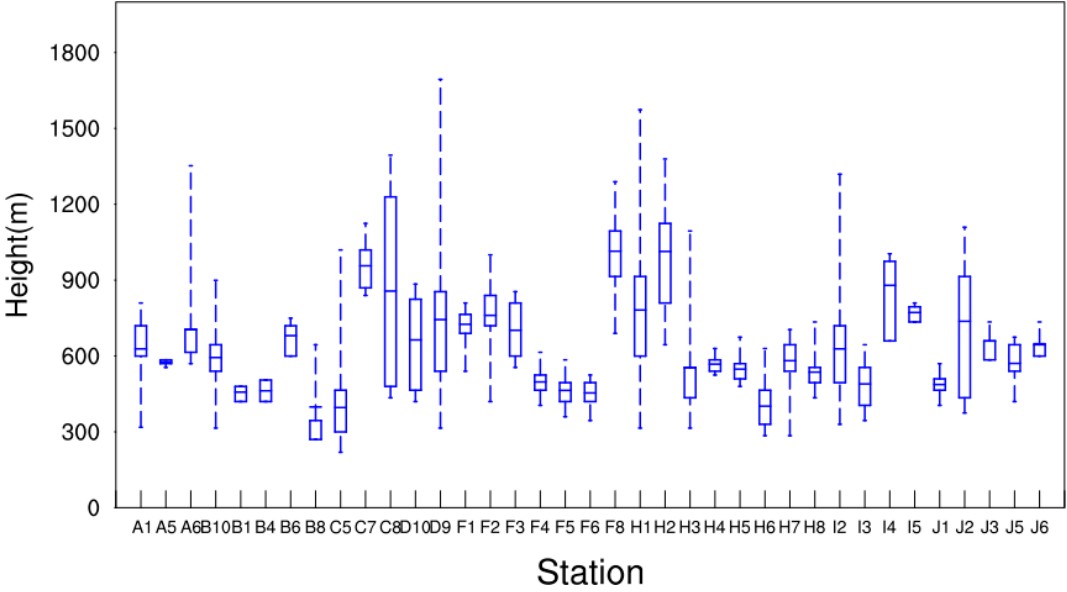

**Figure 7.** The atmospheric boundary layer height calculated along cruise, grouped by stations. Blue boxes show 25th percentile, median, and 75th percentile values. Dashed lines show maximum and minimum values.

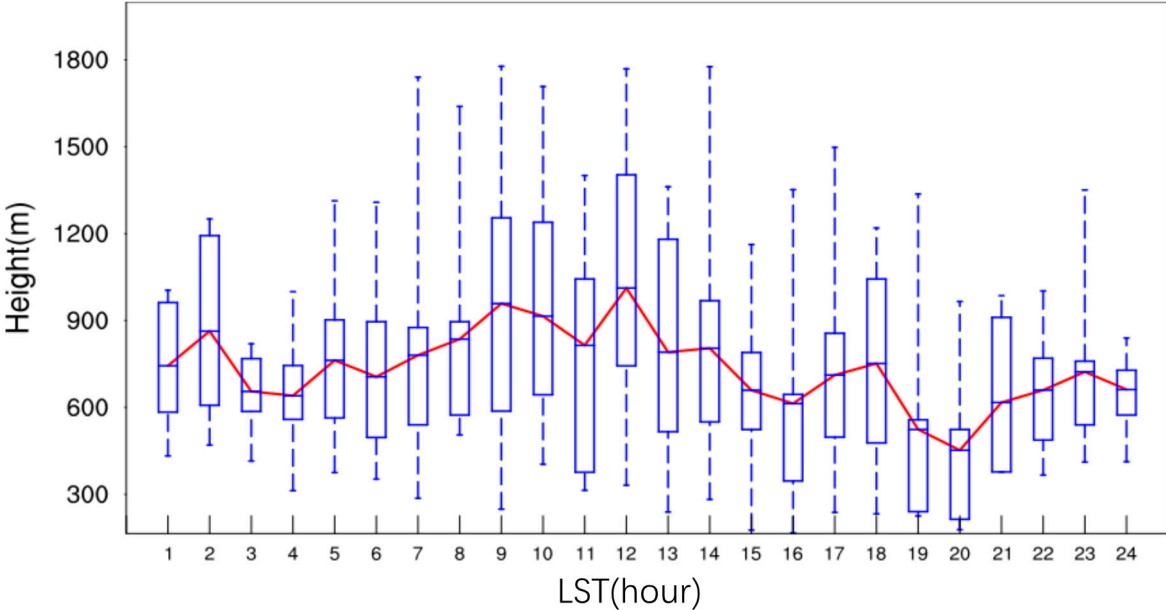

**Figure 8.** The diurnal cycle of atmospheric boundary layer height (ABLH) over the entire cruise. Blue boxes show 25th percentile, median, and 75th percentile values. Dashed lines show maximum and minimum values. The red line shows the mean hourly ABLH. LST means local standard time.

### 3.4. Back-Trajectories

The analyses in the previous sections indicate that terrestrial aerosols were transported into the cruise region, but not uniformly. Figure 9 shows a selection of back-trajectories initiated from measurement stations and times. The authors will limit their analysis below to the northeast region (A–F), avoiding the southwest region where the low-level trough was located (along ~18°N in Figure 3). Furthermore, the discussion below will focus on low-level patterns of aerosol concentrations.

Using extinction coefficient as a proxy for aerosol concentration, the highest concentrations of aerosol were measured below 0.5 km at many stations nearest to the coast: A1 (Figure 5a), B1 (Figure 5b), C5 (Figure 5c), and F1-F2 (Figure 5e). Red lines in Figure 9 show back-trajectories initiated at 200 m over these stations. Back-trajectories from B1 (Figure 9a), C5 (Figure 9b), and F1 (Figure 9c) were similar, all arriving from the southwest. Those from B1 and C5 passed close to the southern coast of Hainan Island, suggesting it to be a source of terrestrial aerosols. In the case of B1, the source may have been more local since the back-trajectory was extremely close to the surface. Back-trajectories from B2-B8 (but not B10, Figure 9e) also arrived from the southwest. In contrast, back-trajectories from F1-F2 remained a distance away from the coast, travelling over the South China Sea from a south-southwest direction.

Unlike these back-trajectories, the back-trajectory initiated at 200 m over A1 arrived from the north instead, as did back-trajectories from A5 and A6 (Figure 9d). Trajectories from A- and B- stations were perturbed. The northern versus southern source of the back-trajectories was robust to perturbations of initiation location and time for A1–A6 and B1–B4. Despite their close physical proximity to one another and elevated aerosol concentrations below 0.5 km, the source of the aerosols appear different. An alternative is that they all shared similar, local sources; knowledge of other aerosol properties at the measurement and coastal locations are needed to make a definite conclusion.

From the above analyses, the authors observed that source history matters more than absolute distance from the nearest coastline in determining near-surface aerosol concentrations, in terms of terrestrial aerosol contamination. For example, C5 was much further away from the coast compared to F1-F2, but aerosol concentrations below 0.5 km over C5 were higher than over F1-F2. The observation will be supported by the further analyses below, showing aerosol concentration generally decreased with absolute distance from the coast, because distance from source, but not always. In the case of section D, the latter was more important.

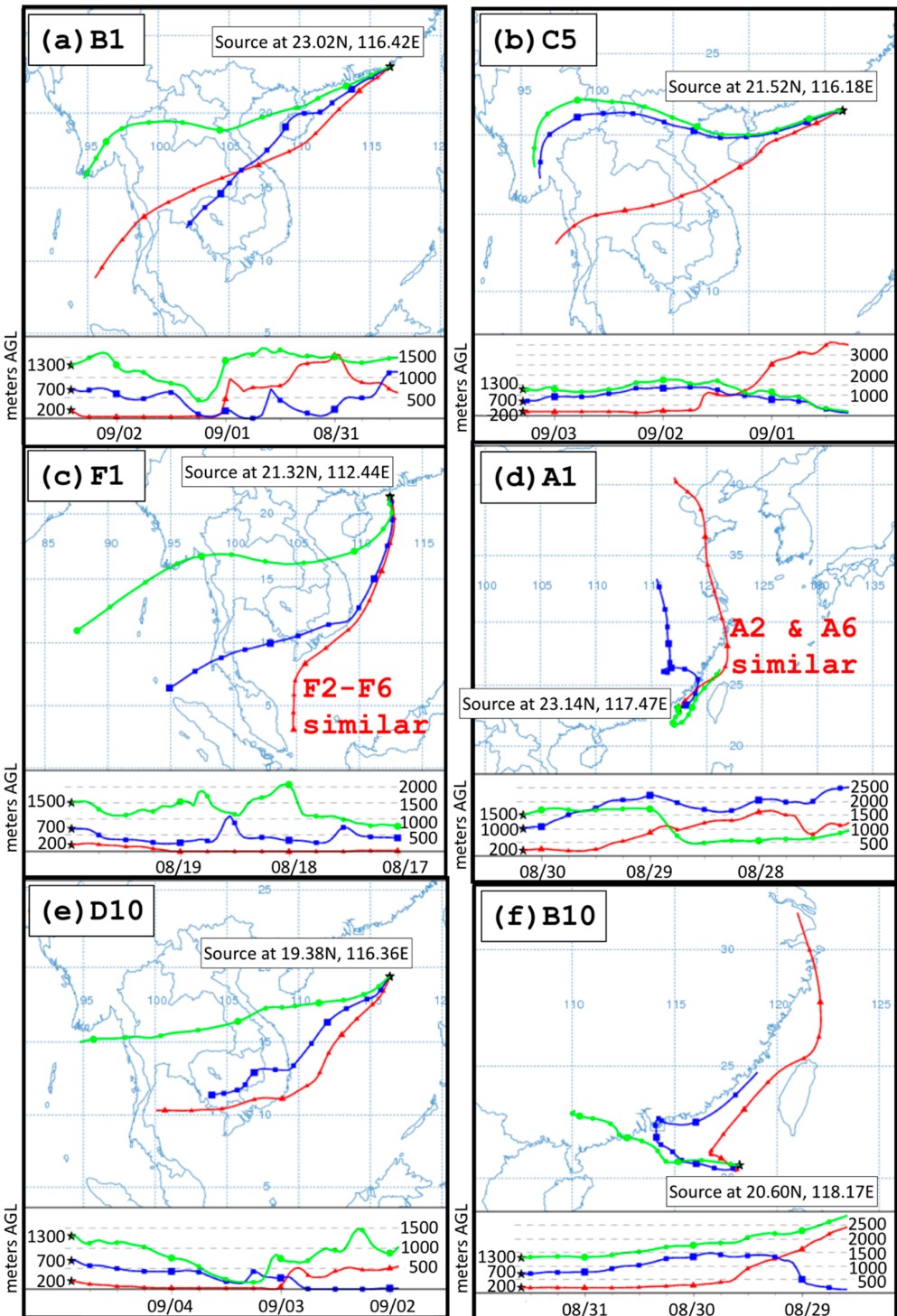

**Figure 9.** Back-trajectories ending at six selected stations on their time of observation, as simulated by HYSPLIT (c.f. Section 2.6): (**a**) B1, (**b**) C5, (**c**) F1, (**d**) A1, (**e**) D10, (**f**) B10. Red lines represent the paths of air parcels initiated at altitudes of 200 m. Blue lines represent the paths of air parcels initiated at 700 m (panels a, b, c, e, f) and 1000 m (panel d). Green lines represent paths of air parcels initiated at 1300 m (panels a, b, e, f) and 1500 m (panels c, d). Colored arrows and text describe similar back-trajectories initiated at the same altitude, but from different stations and observation times. Arrow directions indicate region and direction where the trajectories differed from one another (see Section 3.4 for discussion). TW is the location of Taiwan Island and HN is the location of Hainan Island.

Aerosol concentrations below 0.5 km generally reduced along the sections in the northeast region: A1–A6 (Figure 5a), B1-B8 (Figure 5b), C5–C8 (Figure 5c), and F1–F4 (Figure 5e). D9-D10 were both far from the coast and not considered (Figure 5d). Back-trajectories initiated at 200 m over the A- stations were similar but shifted increasingly eastwards, i.e. away from the coast following A1 to A6 (Figure 9d). Back-trajectories from the B- stations were again similar but shifted increasingly eastwards, i.e., away from Hainan Island and into the South China Sea following B1 to B8 (Figure 9a). Finally, back-trajectories from the F- stations were similar and shifted increasingly eastwards, i.e. away from the coast following F1 to F4 (Figure 9c), though this was only clear when considering the distribution of perturbed back-trajectories. All three sets of back-trajectories (A, B, F) displayed source histories further away from the coast following their initiation distance, which was expected following the seasonal winds over the South China Sea.

Back-trajectories initiated at 200 m over C5 and C7 both crossed Hainan Island along the south (C5) or north (C7), in contrast to the back-trajectory from C8, which traveled across the South China Sea and resembled back-trajectories from the F section (Figure 9b). All three types of back-trajectories were possible in perturbed back-trajectories, with perturbed back-trajectories from C8 strongly favoring the path over the South China Sea. This accounted for the much lower aerosol concentration below 0.5 km at C8 in contrast to C5 and C7.

Section D stood in contrast to sections A, B, C, and F. Although D10 was further from the coast compared to D9, the back-trajectory initiated at 200 m over D10 was further away from the coast compared to that from D9 (Figure 9e), and only intersected the Vietnam coast at the southern tip of Vietnam. This would account for the higher aerosol concentration below 0.5 km at D10 compared to D9. This illustrates the importance of air-parcel source history over coastal distance.

The authors next examined the increase in aerosol concentration below 0.5 km at stations a distance off-shore, i.e. stations B10, D10, and F5-F6. D10 has been accounted for above. The back-trajectory from B10 arrived from the north, following a path close to the coast similar to back-trajectories from the A section, different from other B stations (Figure 9f, compared to Figure 9a,d). Hence, the increase in aerosol concentration off-shore along Section A was likely due to transport from different sources. The authors were unable to account for the increase in F5-F6; back-trajectories from F5 and F6 were very similar to back-trajectories from the other F stations.

Finally, the authors will briefly discuss the possible presence of aerosol layers higher above the surface. Layers of higher aerosol concentration at 0.5–1.0 km and 1.0–1.5 km seemed to exist above the surface at some stations. This was particularly distinct in the vertical profiles of extinction coefficient at A- stations, B4, B10, C5, and F6. Blue and green lines in Figure 9 show back-trajectories initiated at 0.7–1.0 km and 1.3–1.5 km, respectively, matching altitudes with maxima in extinction coefficient.

Over the A- stations and B10, one aerosol layer at 1.0–1.5 km could be seen. Back-trajectories from these stations followed tracks similar to the blue and green lines in Figure 9d,f. The back-trajectories all either arrived from the Fujian (A-) or Guangdong (B10) province, or remained near the local coastal regions.

Unfortunately, the authors were unable to account for the origin of the aerosol layers over the rest of the stations. Over B4, two aerosol layers could be seen at 0.5–1.0 km and 1.0–1.5 km. Back-trajectories from different B stations all followed tracks similar to the blue and green lines in Figure 9a, which passed over Hainan Island. Over C5, one aerosol layer could be seen at 0.5–1.0 km. Back-trajectories from different C stations all followed tracks similar to the blue and green lines in Figure 9b, which again passed over Hainan Island. Over F6, two aerosol layers could be seen at 0.5–1.0 km and 1.3–1.7 km. Moving from F1 to F8, back-trajectories increasingly traveled north into Guangdong first before looping back into the stations (Figure 9c), but the back-trajectory from F6 did not stand out. In summary, the authors could not determine anything unusual about back-trajectories from stations with strong aerosol layers, compared to back-trajectories from stations without such layers. It may even be possible that these layers may have been measurement errors caused by the instability of the cruise vessel.

## 4. Discussion

Extinction coefficients over the northeastern part of the study region were higher than values over the southwestern part. Besides high levels of aerosols near the surface (below 0.5 km) over many stations, additional layers of aerosols were detected at 0.5–1.0 km and 1.0–1.5 km over some stations. The backscatter depolarization ratio indicated that surface and layered regions of higher aerosol concentration were contaminated by terrestrial aerosols, although the aerosol remained in bulk more marine than terrestrial in nature, with a mean depolarization ratio of 0.04 over 0–4 km. The mean atmospheric boundary layer height over the study region was about 653 m with a small diurnal range, typical of marine atmospheric boundary layers.

Back-trajectory analyses supported our hypothesis that terrestrial aerosols were in general advected from the southwestern to the northeastern regions of the study region, by prevailing southwesterly low-level winds of the summer monsoon. For determining aerosol concentration, the source history of the air parcel over any particular spot mattered more than its absolute distance from the nearest coastline. The aerosol source for some locations and vertical layers could have been local from Guangdong, northern from Fujian province, or southwestern from Hainan Island. The distribution and sources of aerosols over the study region appeared complex and layered in both horizontal and vertical dimensions.

Since this was the first time a mini-MPL lidar was used for aerosol observation over the study region, and the first time the authors conducted observations over marine conditions, only 11% of the raw measurements were usable due to a number of technical difficulties. Firstly, the authors had not anticipated the extent of ship instability in the cruise region and had not installed a stabilization rig for the mini-MPL. This lack was exacerbated by the presence of typhoons near the cruise region, which produced large ocean swells and frequent precipitation. Secondly, high humidity in the marine environment resulted in frequent water condensation on the lidar eye lens, greatly reducing the amount of usable measurements. Thirdly, scattering by water vapor (relative humidity > 80%) frequently attenuated the echo signal beyond what was usable.

The authors suspect that the low energy of Mini-MPL as opposed to traditional MPLs may also have limited lidar penetration in the marine condition and led to discontinuities in observation or contamination of the measurements. Research has shown that absorption of the atmospheric gases should be taken into account when processing the return signals of space-based lidars, especially absorption by $O_3$ at the wavelength of 532 nm [51]. Similar effects may have affected this study and in future the absorption of the atmospheric gases should be taken into account in theoretical calculation and the setup of the lidar channel, especially when detecting high altitude.

## 5. Conclusions

A micro-pulse lidar (Mini-MPL) was used for aerosol measurement on a research cruise around the northern region of the South China Sea, from 9 August to 7 September 2016. Extinction coefficients at offshore locations did not exceed 0.15 km$^{-1}$, much smaller than the typical aerosol extinction coefficients over land. Values at more coastal locations did not exceed 0.2 km$^{-1}$. Attenuation below 1 km and 2 km accounted for more than 40% and 74% of total column extinction, respectively. This indicated that aerosols in the study region were mainly concentrated below 2 km, and almost no aerosols were higher than 3.5 km.

Within this 2 km, layers of aerosols were detected at 0.5–1.0 km and 1.0–1.5 km over some stations. The higher backscatter depolarization ratio in these regions suggested contamination by terrestrial aerosols, although the bulk of the aerosol was marine with a mean depolarization ratio of 0.04 over 0–4 km. Back-trajectory analyses supported this hypothesis, with possible source regions being Guangdong, Fujian, or Hainan provinces.

Although Mini-MPL system is easy to use and economical for those with budget constraints, this system may be more suitable for terrestrial use, in accordance to its design. The authors concluded that the default setup of the Mini-MPL was not appropriate for use in the cruising region; a stabilization

rig, instrument protection cover, and anti-fogging precautions are recommended before another attempt. The authors recommend future researchers take into consideration regional marine conditions when preparing for marine observations.

**Author Contributions:** Conceptualization, W.D. and B.W.; data curation, Y.L.; formal analysis, Y.L., S.-Y.L., Z.Z., and Y.W.; funding acquisition, W.D. and B.W.; investigation, Y.L., B.W., Z.Z., and Y.W.; writing, Y.L., S.-Y.L., and B.W. All authors have read and agreed to the published version of the manuscript.

**Funding:** This research was funded by grants from the National Key Research and Development Program of China (2017YFC0209606, 2018YFC1406201, 2016YFA0602700, 2018YFC1506903), the National Key Project of the Ministry of Science and Technology (JFYS2016ZY01002213), and the National Natural Science Foundation of China (91644225, 41775015, 41630422).

**Acknowledgments:** Many thanks to Ruoyu Bao, Sun Yat-sen University, for help installing the lidar and Climate Change and Application Platform. The authors gratefully acknowledge the NOAA Air Resources Laboratory (ARL) for the provision of the HYSPLIT transport and dispersion model and/or READY website (http://www.ready.noaa.gov) used in this publication.

**Conflicts of Interest:** The authors declare no conflict of interest.

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
