# Peer review of "Micro-Pulse Lidar Cruising Measurements in Northern South China Sea"

_remotesensing, doi:10.3390/rs12101695_

Round 1

Reviewer 1 Report

Line 17-18 says extinction was greater over coastal areas and lower over open ocean. But 18-20 say the opposite. Also, you need to say at what wavelength you are talking about (516 nm ?)

Line 19 says the extinction was between 0.016 and 0.039 km-1. If I use an avg value (say 0.025), and assume a polluted layer of 3.5km (as you mention on line 17) then the optical depth is 0.025x3.5 = 0.0875. This corresponds to clean air and not polluted air coming from Asia where the AOD should be between 0.1 to 0.6. I suspect a problem with your derived extinction coefficient numbers.

Line 175-176. The backscatttering coeff is the normalized phase function (at 180 deg scattering angle) times the scattering coeff. The extinction coeff is the scattering coeff plus the absorption coeff. This is not clear in your description.

Lines 248-256 Since you do not have a way to constrain s, it could explain why your lidar derived extinction coefficient values are not reasonable. You really need to read some literature or do some calculations to see what reasonable values are for the aerosol extinction coefficient in that region. Then do a sensitivity study of s to see if you get agreement.

Reviewer 2 Report

My comments were taken into account in corrected version of the manuscript. Paper can be published.

Round 2

Reviewer 1 Report

I have reviewed their response and changes. I suggest the paper is not
excellent but still acceptable for publication. I do not want to review
it again.

The author says they want to calculate the lidar overlap function and
constrain the inversion with a sun photometer. The sun photometer is a good approach. Another approach I suggest is using horizontal lidar measurements. They can look at my horizontal lidar calibration paper online.

Porter, J.N., Lienert, B., S.K. Sharma, Using the Horizontal and Slant Lidar Calibration Methods To Obtain Aerosol Scattering Coefficients From A Coastal Lidar In Hawaii, Journal of Atmospheric and Oceanic Technology, 17, 1445-1454, 2000.

sincerely
John Porter

This manuscript is a resubmission of an earlier submission. The following is a list of the peer review reports and author responses from that submission.

Round 1

Reviewer 1 Report

Abstract

Need to give units for extinction, also is this aerosol only or aerosol + molecular extinction ?

In the abstract you need to say why the instrument did not work well and why you do not recommend it.

Line 84 says the atmospheric extinction coefficient can be measured. Is this the aerosol extinction coefficient? Briefly describe how is that done?

Line 144-146. Can you explain the method you used to derive the aerosol extinction coefficient in words (more detail) to make it more clear.

Line 148 It is important to describe sa in terms of the scattering and absorption coefficients and the aerosol phase function at 180 degrees. When done in this way it becomes clear that the ratio depends on the aerosol absorption vs scattering as well as the aerosol phase function (at 180 degree scattering angle).

Before discussing the results (Section 3) it is necessary to describe the overlap function. The overlap function describes how the lidar detection efficiency increases from near zero at range zero, to 1 at some distance away (many 1-2 km). I see in Figure 2 you are starting at ~250 m range.

In doing the inversion shown in Figure 2, you are using a constant s ratio for all heights. This adds some uncertainty. You need to mention this uncertainty in your results.

Line 412 Your description of why the MPL is not suitable for marine use is very weak. Either strengthen your argument or remove it.

Reviewer 2 Report

Review of the article

Micro-pulse Lidar Cruising Measurements in Northern South China Sea

In the work, the interesting experiment of using of ship-borne micro-pulse lidar to measure aerosol characteristics is presented. The results of the measurements are discussed.

Some comments to the article:

Page 4. The absorption of atmospheric gases did not take into account in the methods, used by the authors to treat the lidar signal. In the simulations [LilinYao, Fu Wang, et al Extinction effects of atmospheric compositions on return signals of space-based lidar from numerical simulation. JQSRT. 2018. 210. 180-188], it was shown that O3 absorption should be taken into account to correct the return signals of space-based lidar operating at the wavelength of 532 nm.

What is the influence of the absorption of the atmospheric gases on the results of the measurements with use of ship-borne lidar at 532 nm? This explanation should be added in the article.

Page 8 and 13. The signs of axes in the Figures 4 and 9 are too small.

The article can be published after small correction. 
